# GeneLLM: Inheriting 1.25% of MoE-LLMs to Build Models of 8% Size that Retain 80% Performance

## Abstract

Despite the rapid progress of large language models (LLMs), their enormous training costs and deployment challenges remain significant limitations. To address these issues, the Learngene framework was proposed, offering a one-time extraction of a compact, reusable knowledge module from a large model, which enables smaller models of various sizes to generalize more effectively across downstream tasks. In this work, we are the first to apply the Learngene framework to LLMs. We propose **GeneLLM**, a low-cost initialization framework that allows rapid construction of lightweight domain-specialized models under limited data resources. By performing only forward inference on Mixture-of-Experts (MoE) LLM with multi-domain data, we identify experts of LLM with cross-domain common knowledge and apply tensor decomposition to refine a compact knowledge module, termed learngene. This module is then used to reconstruct and initialize the Feed-Forward Network (FFN) layers of small dense models, enabling efficient adaptation to downstream tasks with minimal computational cost. Experiments show that by inheriting only 1.25% parameters of the source LLM, GeneLLM enables target lightweight models, only 8.1% the size of the source LLM, to recover over 80% of its performance on certain tasks. Compared to training from scratch and distillation, GeneLLM consistently delivers superior task performance, cost efficiency, and faster convergence speed. The code is available at https://anonymous.4open.science/r/GeneLLM-main-0EB3/.

## 1 Introduction

With the rapid advancement of large language models (LLMs)(Wiggins & Tejani, 2022; Chowdhery et al., 2023; Achiam et al., 2023; Grattafiori et al., 2024; Zhong et al., 2025), many models have shown strong performance in various tasks and domains(Zhang et al., 2024; Sun et al., 2024a; Yi et al., 2024). While early LLMs were mainly based on dense architectures (Touvron et al., 2023; Mann et al., 2020; Bai et al., 2023), recent research has shifted toward sparse Mixture-of-Experts (MoE) models (Shazeer et al., 2017; Dai et al., 2024; Muennighoff et al., 2025; Team et al., 2024; Liu et al., 2024a), which scale efficiently by activating only a few specialized experts per input, thereby reducing training and inference costs while achieving superior performance. However, even MoE-based LLMs still contain tens to hundreds of billions of parameters, making deployment challenging, particularly in resource-constrained environments where applications often require only partial functionality (Zhu et al., 2024; Tambe et al., 2021; Gu et al., 2024). This motivates the development of smaller, task-specific models tailored to different computational budgets, for which the Learngene framework (Wang et al., 2023a; Shi et al., 2024) has been proposed to extract compact and knowledge-rich parameter modules from large models to initialize lightweight models across different scales for efficient adaptation.

The idea of Learngene offers a promising solution to the growing gap between the capabilities of large models and the practicality of deploying them. The Learngene framework includes two phases: identifying and extracting essential knowledge module as learngene, and inheriting it into downstream models to enhance learning efficiency. As shown in Figure 1, the large model is typically referred to as a source model from which a compact parameter subset, termed learngene, is extracted. A learngene can correspond to an entire block (Wang et al., 2023a) or a specific parameter matrix within a block of the source model (Xia et al., 2024). The smaller models initialized

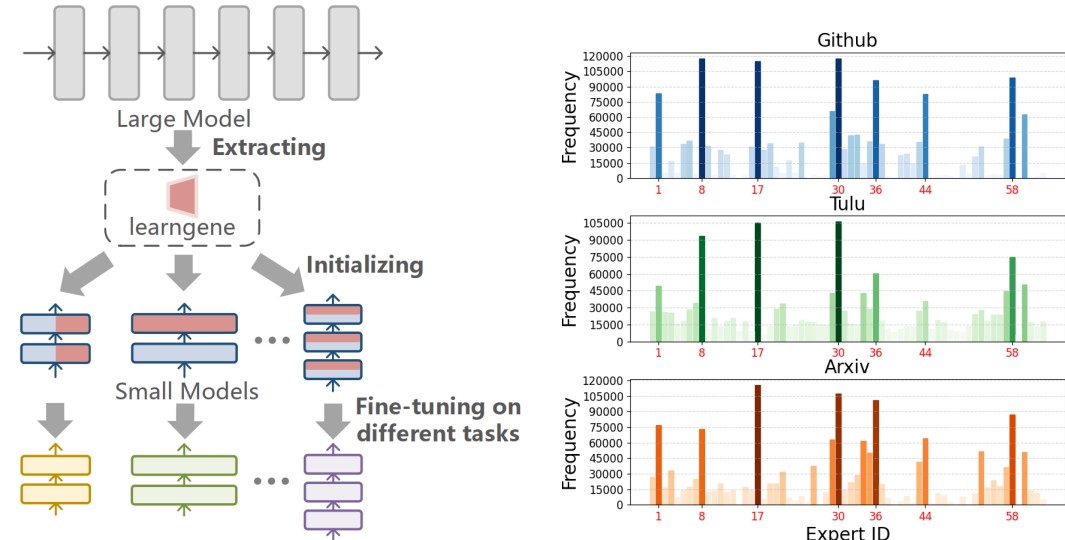

Figure 1: An overview of Learngene framework.

Figure 2: Experts activation frequencies of layer 15 in OLMoE-7B across different domains.

with these learngene can vary in size and be tailored to different downstream tasks. By extracting a compact set of parameters that encapsulate core knowledge from the large model, learngene enables the initialization of smaller task-specific models that can generalize quickly with minimal training. Compared to traditional compression methods such as distillation (Gu et al., 2024), which require repeated forward passes through the teacher model, Learngene extracts reusable components from the source large model just once and reuses them in target lightweight models of varying sizes and tasks, offering greater flexibility and efficiency.

However, while this paradigm has shown promising results in vision tasks and small-scale settings (Wang et al., 2023a; Xia et al., 2024; Shi et al., 2024), it has not yet been explored in the context of LLMs. In this work, we take the first step in extending Learngene to LLMs by proposing a novel initialization framework, **GeneLLM**. Our method is designed to be highly cost-efficient: (i) it requires no additional training during the extraction stage, (ii) avoids repeated forward inference on the source model, and (iii) relies on substantially less pre-training data compared to random initialization. By selectively extracting **cross-domain common knowledge** from the source LLM in a one-time process, GeneLLM enables target lightweight models of different scales to quickly adapt to **a wide range of downstream sub-tasks**.

GeneLLM is an initialization framework that leverages MoE-based LLMs as knowledge sources to build lightweight models. Compared to dense LLMs where knowledge is entangled across the entire parameter space, MoE LLMs localize knowledge through expert routing, enabling clearer attribution of capabilities to specific experts. This property makes MoE a natural foundation for extracting the learngene. A key question, then, is how to identify experts that encode broadly generalizable knowledge. An intuitive approach, also explored in previous work on expert activation analysis (Fedus et al., 2022; Huang et al., 2024), is to run forward passes with data from different domains and observe which experts are consistently activated. To ensure these experts indeed capture domain-agnostic knowledge, the input domains should be both diverse and representative of common pre-training sources. Since large-scale pre-training typically draws from web text, code, and scientific literature, we use Github, Tulu, and arXiv as domain-specific inputs. As shown in Figure 2 (see more details in A.1), in OLMoE-7B certain experts (e.g., 1, 8 and 17 in layer 15) are repeatedly activated across these domains, suggesting that they encode shared knowledge among domains. We regard such experts, which remain highly active across diverse domains, as carriers of common knowledge. Similarly, in architectures (DeepSeekMoE) with shared experts that are activated across all tasks, we directly treat these shared experts as experts with common knowledge.

In GeneLLM, as shown in Step 1 of Figure 3, we identify experts in the source LLM by performing only forward inference with data from multiple domains, without any additional training or fine-tuning. This purely inference-based process is lightweight, incurs minimal computational overhead, and efficiently selects experts that consistently capture cross-domain common knowledge.

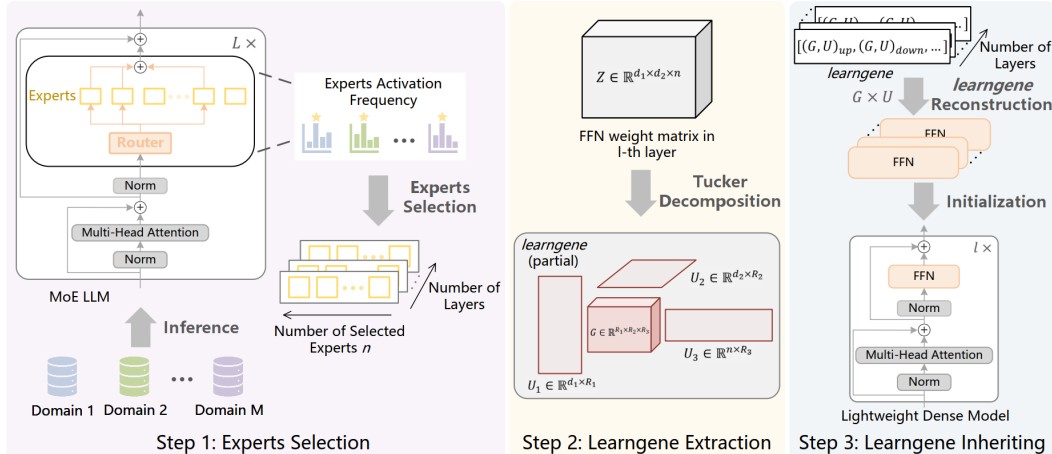

Figure 3: The framework of GeneLLM. $Z$ is the tensor formed by stacking the selected experts parameter matrices, and $(\mathcal{G}, U)$ represents the learngene corresponding to a specific parameter matrix.

To further purify this knowledge into a compact and generalizable form, we introduce a purification step. As described in Step 2 of Figure 3, the selected experts from each block are stacked into high-order tensors, upon which we apply tensor decomposition. This decomposition extracts the learngene, a compact module that retains the essential domain-agnostic knowledge embedded across experts. Finally, since target models are typically small in size and designed for a single downstream task, we adopt a dense model architecture for our target small models. As illustrated in Step 3 of Figure 3, the learngene for each layer is reconstructed into Feed-Forward Network (FFN) parameter matrices that match the dimensionality of the target small dense model and are directly used to initialize its FFN layers, while the remaining parameters are randomly initialized.

In our experiments, we use OLMoE-7B (Muennighoff et al., 2025) and DeepSeekMoE-16B (Dai et al., 2024) as the source LLMs. To thoroughly evaluate GeneLLM, we select supervised fine-tuned (SFT) datasets from two task types, language understanding and dialogue generation, with each task type further encompassing multiple vertical domains. The extracted learngene accounts for only **1.25%** of source LLM's parameters, and the resulting lightweight models range **from 2.4% to 8.1%** of the source models' size. Remarkably, with pre-training on only 2B–5B tokens data, GeneLLM achieves over **80%** of the fully fine-tuned LLM's performance on certain SFT tasks. These lightweight models can be effectively fine-tuned and, in some cases, even outperform scratch-trained baselines while requiring **5× less** pre-training data. Extensive experiments show that learngene-initialized models consistently outperform baselines such as training from scratch and distillation, while converging significantly faster. Overall, these results demonstrate the effectiveness, efficiency, and generalization of GeneLLM, and establish it as an low-cost and efficient initialization method for rapidly building domain-specialized lightweight models under limited data resources.

## 2 RELATED WORK

**Model Compression for LLMs.** The rapid growth of LLMs has boosted NLP performance but also raised deployment challenges due to their massive size and computational cost. To address this, three main families of compression methods have been explored: knowledge distillation, quantization, and pruning. Knowledge distillation transfers knowledge from a large teacher to a smaller student model, with techniques like MiniLLM (Gu et al., 2024) leveraging reverse KL divergence for better distribution matching, and approaches such as in-context learning (Huang et al., 2022) and chain-of-thought distillation (Wei et al., 2022; Wang et al., 2023b) enhancing reasoning and task performance. Quantization reduces precision, either during training (QAT (Liu et al., 2023; Du et al., 2024; Xu et al., 2024)) or post-training (PTQ (Park et al., 2024; Dettmers et al., 2022; Frantar et al., 2023; Lin et al., 2024)), with the former enabling adaptation to low-bit formats and the latter preserving model structure without retraining. Pruning eliminates redundancy by removing weights (unstructured pruning (Frantar & Alistarh, 2023; Sun et al., 2024b; Xia et al., 2023)) or larger units such as layers and heads (structured pruning (Ma et al., 2023; Kim et al., 2024; Ashkboos et al., 2024)), while maintaining model integrity.

**Learngene.** The Learngene framework, first proposed by Auto-Learngene (Wang et al., 2023a), identifies inheritable knowledge ("learngene") from a large source model through meta-learning and similarity comparisons between layers of pseudo descendant models and the original network. TLEG (Xia et al., 2024) further revealed linear correlations among Vision Transformer (ViT) blocks, enabling parameter-sharing linear combinations to construct compact learngenes with reduced training costs. Learngene Pool (Shi et al., 2024) introduced an approach using SN-net to distill knowledge into auxiliary models, which are then stitched into smaller architectures of various sizes. A related concept, Learnware (Zhou & Tan, 2023), also promotes reusable modular knowledge but differs fundamentally: Learngene extracts a shared, compact representation from one large model, while Learnware trains separate task-specific modules for dynamic inference-time composition.

**Comparison of Learngene and LLMs Compression.** Traditional compression methods often suffer performance collapse when there is a large capacity gap between the original and target models (Mirzadeh et al., 2020; Movva et al., 2022). For instance, in knowledge distillation, a significant size gap between the teacher and student models can make it difficult for the student to effectively learn from the teacher, leading to poor distillation performance (Mirzadeh et al., 2020; Gao et al., 2020; Wang & Yoon, 2021). Even the most effective quantization techniques typically compress the model to no less than 10% of its original size (Xu et al., 2024). However, in GeneLLM, the size of the smallest target model is only **2.44%** of source LLM, which means that the source LLM is **40×** larger than the target lightweight model. Moreover, traditional compression methods require repeated costly forward passes through large models to produce variants for different sizes or tasks. In contrast, Learngene paradigm extracts reusable components once, enabling efficient initialization of diverse lightweight models while improving generalization and reducing training costs.

## 3 METHOD

As previously described, GeneLLM consists of two main steps: extracting learngene from the MoE LLMs and reconstructing them to initialize lightweight models of various scales. Specifically, we introduce how to select experts from source LLM for learngene extraction in Section 3.2, and then explain the extraction method in Section 3.3. After extracting, in Section 3.4 we show how the extracted learngene is reconstructed and applied to the initialization of target lightweight models.

### 3.1 PRELIMINARY

The MoE architecture consists of multiple experts $E$ and a router $G(x)$. The input data are processed through the router, which selects a subset of experts for computation. In an MoE layer with $N$ experts, each token $x$ is assigned a subset of experts for computation through the router:

$$y = \sum_{i=1}^{N} G_i(x) E_i(x),$$ (1)

$$G_i(x) = \begin{cases} g_i, & \text{if } g_i \in TopK(\{g_j \mid 1 \leq j \leq N\}, \ K), \\ 0, & \text{otherwise}, \end{cases}$$ (2)

where $E_i$ represents the $i$-th expert, which is implemented as the FFN layer in a Transformer model. The term $g_i = \text{softmax}_i(Wx)$ denotes the softmax value obtained for the $i$-th expert after the gating mechanism processes the input $x$. $W$ represents the weight parameters of the gating function.

In some recent MoE models (Dai et al., 2024; Liu et al., 2024a), shared experts have been introduced to handle common knowledge in different tasks. The calculation of an MoE architecture with shared experts is formulated as $y = \sum_{i=1}^{N} G_i(x) E_i(x) + \sum_{i=1}^{S} E_i^s(x)$, where $E^s$ represents the shared experts, $S$ is the number of the shared experts.

### 3.2 EXPERTS SELECTION

As previously discussed in the Introduction, we refer to experts that exhibit consistently high activation across multiple domains as experts with common knowledge, as they are likely to capture knowledge shared across domains. Since our goal is to extract the learngene that encapsulates common knowledge, the first step is to identify experts with common knowledge. In certain MoE designs, such as DeepSeekMoE, shared experts are explicitly designated to capture such common knowledge, and we directly utilize them as experts with common knowledge. However, in more general MoE settings where shared experts are not explicitly defined, we propose a method to automatically select experts that are likely to encode common knowledge.

As shown in Step 1 of Figure 3, we begin by utilizing $M$ different domain-specific datasets $D_1, D_2, \ldots, D_M$, each corresponding to a different domain. We define the activation probability of the expert $i$ on the dataset $D_m$ as $P_{i,m}$. $P_{i,m}$ reflects the usage probability of the $i$-th expert on domain $m$, that is, the proportion of the number of times the expert $i$ is activated to the total number of activation of all experts.

To identify experts that are active across multiple domains, we compute the average activation level of expert $i$ across all datasets:

$$A_i = \frac{1}{M} \sum_{m=1}^{M} P_{i,m}. \tag{3}$$

In the above function, a higher $A_i$ indicates that expert $i$ is frequently activated across multiple domains, suggesting that it encapsulates a greater amount of shared knowledge among domains. Furthermore, to avoid selecting experts that exhibit high activation in only a few domains while remaining inactive in others, we also consider the consistency score, which measures the balance of an expert's activation across different domains. A lower consistency score of expert $i$ implies that the activation of expert $i$ is relatively balanced across all the domains, rather than being highly responsive to only a few domains. The consistency score of the expert $i$ is:

$$C_i = \frac{1}{M} \sum_{m=1}^{M} |P_{i,m} - A_i|. \tag{4}$$

Finally, by combining the average activation level of experts across all datasets with their activation consistency, we identify the experts with cross-domain common knowledge: $\varepsilon = TopK(\{A_i - C_i | 1 \le i \le N\}, \ n)$, where $n$ represents the number of experts with common knowledge to be selected. $\varepsilon$ is the set of the selected experts with common knowledge.

## 3.3 LEARNGENE EXTRACTION

While the experts we selected are good at handling common knowledge across domains, the shared knowledge they contain still requires further refinement. To preserve higher-purity shared knowledge from these experts, we need to keep the common information between them and remove their differences, which helps create cleaner shared knowledge.

Specifically, we adopt the Tucker decomposition (Tucker, 1966) of high-dimensional tensor to extract learngene from the selected experts, aiming to purify shared knowledge. Tucker decomposition factorizes a high-dimensional tensor into a core tensor and several factor matrices, retaining only partial information along each dimension. The core tensor obtained through Tucker decomposition captures the relationships between the principal components across different dimensions of the original tensor. By retaining only a small number of principal components, the impact of noise from individual experts is reduced, allowing the core tensor to effectively capture key commonalities among experts (more detailed analysis is given in the Appendix A.6).

First, we stack the expert matrices together to form a higher-order tensor $Z \in \mathbb{R}^{d_1 \times d_2 \times n}$. Here, $d_1$ and $d_2$ are the dimensions of the expert matrices from the LLM. For example, $d_1$ could be the embedding dimension in the FFN layer, and $d_2$ might be the intermediate size in the same FFN layer. Here, we only describe the case where the target embedding dimension and FFN intermediate size of the target lightweight model are consistent with those of the source LLM. Therefore, $d_1$ and $d_2$ also correspond to the dimensions of the expert matrices in the lightweight model. We provide a detailed explanation in the Appendix A.5 on how to adjust the process when the target lightweight model size differs from that of the source LLM.

Step 2 in Figure 3 provides a detailed illustration of the process. To further extract common knowledge from selected experts while simultaneously reducing the scale of the learning components, we employ the Tucker decomposition to purify the learngene from the higher-order tensor $Z$:

$$\mathcal{G}, \ U = Tucker(Z, \ (R_1, \ R_2, \ R_3)), \tag{5}$$

where $\mathcal{G} \in \mathbb{R}^{R_1 \times R_2 \times R_3}$ represents the core tensor, $U = (U_1, \ U_2, \ U_3)$ and $U_i$ denotes the factor matrix along the $i$-th dimension. The values $R_1$, $R_2$, and $R_3$ are hyperparameters that determine the rank of the principal components retained in each dimension. The combination of the core tensor $\mathcal{G}$ and the factor matrices $U$ constitutes the learngene extracted from selected experts. The extracted learngene, obtained through the above method, can be as small as approximately **1.25%** of the total parameters of the source LLM (e.g., OLMoE-7B).

Table 1: Results of models with different scales on language understanding benchmarks. To evaluate models of different scales, we experiment with 8, 12, and 16 layers lightweight models. All baselines are first pre-trained on 5B tokens, and then fine-tuned for 3 epochs on these datasets.

| Model Params | Baseline | Commonsense & Reading Comprehension | | | | Education | Law | Medicine | Avg. |
|---|---|---|---|---|---|---|---|---|---|
| | | BoolQ | Hellaswag | WinoGrande | PIQA | MMLU | CaseHold | MedMCQA | |
| 8 Layers 391M | Scratch | 70.83 | 26.95 | 49.09 | 50.00 | 28.74 | 78.50 | 32.09 | 48.03 |
| | Distillation | 71.32 | 26.21 | 49.41 | 50.22 | 27.83 | 80.48 | 32.30 | 48.25 |
| | Pruning-EEP | 64.10 | 25.40 | 48.62 | 48.59 | 23.38 | 61.74 | 30.22 | 43.15 |
| | GeneLLM-OLMoE | **72.63** | **27.27** | **50.59** | 51.80 | 29.92 | **81.50** | 34.16 | 49.70 |
| | GeneLLM-DeepSeek | 71.56 | 27.15 | 50.36 | **52.77** | **30.44** | 81.10 | **34.50** | **49.70** |
| 12 Layers 480M | Scratch | 71.56 | 26.80 | 48.86 | 51.90 | 29.07 | 80.97 | 32.78 | 48.85 |
| | Distillation | 71.62 | 26.40 | 49.17 | 50.87 | 28.02 | 81.35 | 32.39 | 48.55 |
| | Pruning-EEP | 65.50 | 25.61 | 48.54 | 48.91 | 23.91 | 60.37 | 27.95 | 42.97 |
| | GeneLLM-OLMoE | **73.76** | **27.40** | **50.36** | **55.28** | 30.44 | **82.90** | 35.04 | **50.74** |
| | GeneLLM-DeepSeek | 72.35 | 26.47 | 50.12 | 51.25 | **31.61** | 82.12 | **35.50** | 49.92 |
| 16 Layers 570M | Scratch | 70.86 | 26.29 | 48.62 | 51.89 | 29.50 | 80.16 | 33.04 | 48.62 |
| | Distillation | 71.77 | 26.47 | 48.86 | 52.68 | 28.49 | 81.35 | 33.52 | 49.02 |
| | Pruning-EEP | 63.82 | 25.72 | 48.38 | 50.54 | 24.62 | 62.37 | 27.71 | 43.31 |
| | GeneLLM-OLMoE | **73.91** | **27.56** | 49.96 | **55.44** | **33.05** | **82.96** | **34.78** | **51.09** |
| | GeneLLM-DeepSeek | 72.14 | 27.44 | **50.75** | 54.84 | 30.31 | 82.54 | 34.47 | 50.36 |

### 3.4 Learngene Reconstruction and Inheriting

We provide a detailed description of this part in Step 3 of Figure 3. After extracting and refining the learngene from the selected experts, we need to reconstruct it into lightweight models to enable adaptation across diverse downstream tasks. Based on the previous extraction step, we have obtained the core tensor $\mathcal{G} \in \mathbb{R}^{R_1 \times R_2 \times R_3}$ and the factor matrices $U$. In this step, we first apply an averaging operation to the $U_3 \in \mathbb{R}^{R_3 \times n}$ factor matrix $\bar{U}_3 = 1/n \sum_{i=1}^{n} U_3[:, i]$. The dimension of $\bar{U}_3$ is $R_3 \times 1$. Then we perform a dimension-wise multiplication between the core tensor and the factor matrices:

$$\Gamma = \mathcal{G} \times_1 U_1 \times_2 U_2 \times_3 \bar{U}_3, \tag{6}$$

where $\Gamma \in \mathbb{R}^{d_1 \times d_2}$ is the parameter matrix in FFN layer. We use $\Gamma_{l,j}$ to represent the $j$-th parameter matrix in the FFN layer of the $l$-th block, which is used to initialize the corresponding part of the lightweight model. By initializing the FFN layers of a dense model with the corresponding block's $\Gamma$ matrix, we obtain a lightweight model that inherits the learngene.

In lightweight dense models, only the FFN components of each block are initialized with the reconstructed learngene, while all other parameters are randomly initialized. Following this initialization, lightweight models can achieve rapid convergence and exhibit strong generalization in both language understanding and generation downstream sub-tasks even with limited pre-training data.

## 4 Experiments

Our goal is to extract a compact and generalizable learngene from LLMs to empower smaller ones with rapid and efficient adaptation across sub-tasks. Accordingly, we compare GeneLLM with lightweight baselines across two dimensions: **task type**, including language understanding and dialogue generation, and **domain specialization**, covering diverse vertical knowledge domains.

### 4.1 Baselines and Datasets

**Baselines.** We set up the following baselines for comparison experiments. **Scratch:** model initialized from scratch. **Distillation:** model pre-trained with knowledge distillation from the LLM, starting from random initialization. **Pruning-EEP:** model obtained by applying the experts pruning based method EEP Liu et al. (2024b) to the MoE-based LLM (OLMoE). **GeneLLM-OLMoE:** Lightweight model initialized with the reconstructed learngene extracted from the LLM OLMoE-7B (Muennighoff et al., 2025). **GeneLLM-DeepSeek:** Lightweight model initialized with the reconstructed learngene extracted from the LLM DeepSeekMoE-16B-base (Dai et al., 2024).

All baselines are implemented using the same dense model architecture. To ensure fair comparisons, models of the same scale have an equal number of parameters across different baselines and all baselines undergo pre-training with the same amount of tokens. Following pre-training, the baselines are fine-tuned separately on each SFT dataset, allowing evaluation of their performance on each single

Table 2: Results of models with different scales on dialogue generation benchmarks. For evaluating generative capabilities of models, models (pre-trained on 5B tokens) are first fine-tuned on the Dolly dataset for 100 epochs, and then evaluated on the datasets shown in the table.

| Model Params | Baseline | Dialogue Generation | | | | | Avg. (Rouge-L) |
| --- | --- | --- | --- | --- | --- | --- | --- |
| | | DollyEval | S-NI | UnNI | SelfInst | VicunaEval | |
| 8 Layers 391M | Scratch | 22.94 | 15.69 | 18.16 | 8.50 | 14.25 | 15.91 |
| | Distillation | 22.78 | 16.63 | 18.14 | 8.34 | 13.01 | 15.78 |
| | Pruning-EEP | 13.61 | 8.05 | 7.82 | 5.87 | 11.89 | 9.45 |
| | GeneLLM-OLMoE | 23.17 | **17.39** | **21.47** | **9.18** | **15.05** | **17.25** |
| | GeneLLM-DeepSeek | **23.35** | 17.34 | 20.02 | 8.72 | 13.99 | 16.68 |
| 12 Layers 480M | Scratch | 22.48 | 16.57 | 20.76 | 9.41 | 14.20 | 16.68 |
| | Distillation | 22.90 | 15.70 | 21.47 | 9.01 | 14.39 | 16.69 |
| | Pruning-EEP | 15.53 | 7.78 | 8.78 | 5.66 | 11.77 | 9.90 |
| | GeneLLM-OLMoE | 23.36 | **19.43** | **21.83** | 9.57 | **14.83** | **17.80** |
| | GeneLLM-DeepSeek | **23.38** | 18.01 | 20.48 | **10.30** | 13.92 | 17.22 |
| 16 Layers 570M | Scratch | 22.86 | 15.61 | 19.67 | 8.31 | 13.42 | 15.97 |
| | Distillation | 22.09 | 16.06 | 20.27 | 10.20 | 14.09 | 16.54 |
| | Pruning-EEP | 18.59 | 9.60 | 11.41 | 6.44 | 11.80 | 11.57 |
| | GeneLLM-OLMoE | **24.19** | **19.82** | 22.60 | **11.31** | 14.57 | **18.50** |
| | GeneLLM-DeepSeek | 23.42 | 17.99 | **23.20** | 10.43 | 14.37 | 17.88 |

task-specific dataset. In addition, we include the source LLM as an upper-bound baseline for comparison. According to the results reported in the OLMoE-7B paper, OLMoE-7B consistently outperforms DeepSeekMoE-16B on evaluation tasks. Combined with resource constraints, we therefore choose OLMoE-7B as the upper bound for comparison in our experiments. Specifically, we use the original **OLMoE** without fine-tuning, the fully fine-tuned **OLMoE-SFT**, and the **OLMoE-PEFT** fine-tuned using parameter-efficient fine-tuning (PEFT) as baseline models.

**Datasets.** We fine-tune the target lightweight models on diverse sub-tasks, which are defined along two dimensions: task type (language understanding and dialogue generation) and vertical knowledge domain (e.g., physics, law, medicine). For the understanding sub-tasks, we adopt datasets including **BoolQ** (Clark et al., 2019), **Hellaswag** (Zellers et al., 2019), **MMLU** (Hendrycks et al., 2021), **PIQA** Bisk et al. (2020), **WinoGrande** (Sakaguchi et al., 2021), **CaseHold** (Zheng et al., 2021), and **MedMCQA** (Pal et al., 2022), which are widely used to evaluate LLM comprehension across different domains. For the generative sub-tasks, we follow the evaluation setup of MiniLLM (Gu et al., 2024) and test on **Dolly**[1] , **S-NI** (Wang et al., 2022), **UnNI** (Honovich et al., 2023), **SelfInst** (Wang et al., 2023c), and **VicunaEval** (Chiang et al., 2023).

### 4.2 MAIN RESULTS

To comprehensively evaluate the effectiveness of GeneLLM, we conduct experiments from three key perspectives: **cost efficiency**, **benchmark performance**, and **fine-tuning convergence**.

**Significant savings in pre-training data and computational cost.** We give the fine-tuning results of the 16-layer lightweight models after pre-training on different amounts of tokens in Figure 4. The experimental results clearly show that on the UnNI dataset, even a Scratch model pre-trained with 10B tokens fails to outperform GeneLLM-OLMoE trained with only 2B tokens, demonstrating that GeneLLM achieves significantly better performance while using at least **5×** less pre-training data (see more demonstrations in Appendix A.7). These results highlight the data efficiency and deployment potential of GeneLLM.

In addition, pre-training a 570M model on 5B tokens with GeneLLM requires only 72 GPU hours, compared to 252 GPU hours for distillation, which relies on repeated forward passes through the teacher model. Scaling to $n$ lightweight models adds only $72 \times n$ GPU hours, while distillation scales to $252 \times n$ GPU hours, and further increases when applied across multiple downstream tasks. In general, GeneLLM achieves at least a **3.5×** efficiency gain, highlighting its nature as a highly low-cost initialization method.

---

[1]https://github.com/databrickslabs/dolly/tree/master

Table 3: Results of source LLM and target lightweight model on different datasets.

| Baseline | #Params | DollyEval | S-NI | UnNI | SelfInst | VicunaEval |
|---|---|---|---|---|---|---|
| OLMoE | 7B | 10.78 | 11.10 | 12.25 | 8.46 | 14.42 |
| OLMoE-SFT | 7B | 29.13 | 30.49 | 33.08 | 15.36 | 16.83 |
| OLMoE-PEFT | 7B | 20.06 | 31.96 | 25.90 | 12.24 | 15.37 |
| GeneLLM-OLMoE | 570M | 24.19 | 19.82 | 22.60 | 11.31 | 14.57 |

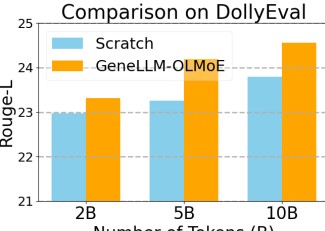 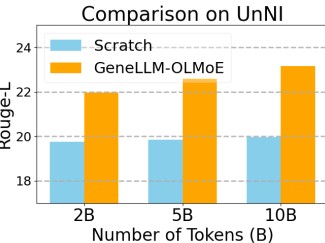 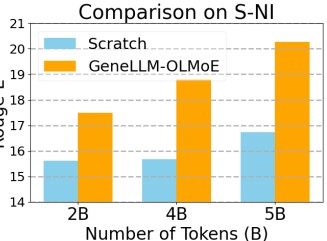

Figure 4: Comparison of fine-tuning performance between 16-layer GeneLLM-OLMoE and Scratch under varying pre-training token budgets.

**Superior performance across benchmarks.** Table 1 and Table 2 report our results on language understanding and generation benchmarks, respectively. As shown in tables, our model consistently outperforms the Scratch, Distillation and Pruning baselines (see more demonstrations in Appendix A.3). The lightweight models initialized with learngene achieve superior performance across nearly all tasks. For example, on generation task S-NI, 16-layer GeneLLM exceeds Scratch by **4.21**, Distillation by **3.76**, and Pruning-EEP by **10.22**. We attribute Distillation's poor performance to the large capacity gap (over $10\times$) between teacher and student models, which hinders effective knowledge transfer (Mirzadeh et al., 2020; Gao et al., 2020; Wang & Yoon, 2021). Similarly, pruning struggles in this setting, as the large model size disparity causes structural damage and disrupts the original functionality. In contrast, GeneLLM mitigates these issues by extracting and reconstructing a compact learngene from the LLM, enabling lightweight models to generalize well across tasks. These results highlight the transferability and generalizability of learngene, demonstrating its effectiveness in bridging the capacity gap while preserving essential knowledge.

**Faster convergence speed.** Figure 5 presents the Rouge-L and loss curves of the 16-layer model Scratch and GeneLLM-OLMoE on the Dolly dataset across epochs. GeneLLM-OLMoE reaches the same Rouge-L score **2.4×** faster than Scratch and achieves the same loss value **1.7×** faster. These results demonstrate that models inheriting the learngene converge more efficiently on downstream tasks, indicating better task adaptability and generalization. Consequently, GeneLLM also reduces the training cost on downstream datasets. The faster convergence demonstrates the strong generalization of learngene, enabling efficient learning with fewer training steps.

### 4.3 COMPARISON WITH SOURCE LLM

Table 3 presents the performance gap between the target lightweight model and source LLM. GeneLLM-OLMoE is a 16-layer, 570M-parameter model pre-trained on only 5B tokens. GeneLLM-OLMoE achieves over **80%** of the OLMoE-SFT performance on several benchmarks. For instance, on DollyEval and VicunaEval, GeneLLM-OLMoE reaches **83.0%** and **86.6%** of the OLMoE-SFT scores, respectively. Compared to OLMoE-PEFT, GeneLLM demonstrates comparable or even superior performance across most tasks. In particular, on DollyEval, GeneLLM-OLMoE outperforms OLMoE-PEFT by a large margin (**+4.13**).

The relatively lower scores on tasks like S-NI and UnNI may stem from higher complexity or annotation noise in these datasets, which can hinder small models from fully leveraging supervision signals. In contrast, on relatively cleaner or more general benchmarks like DollyEval and SelfInst, the general knowledge encoded in learngene can be more effectively utilized, leading to superior results. Overall, these results confirm that learngene enables compact models to inherit broad capabilities from LLMs, achieving strong performance with better efficiency and generalization.

### 4.4 ABLATION ANALYSIS

We evaluate the impact of expert selection and Tucker decomposition–based learngene extraction on GeneLLM. We also analyze how Tucker rank settings affect downstream performance (see Ap-

Table 4: Ablation results for different modules in GeneLLM.

| Method | BoolQ | Hellaswag | MMLU | PIQA | WinoGrande |
|---|---|---|---|---|---|
| w.o. *Experts Selection* | 71.89 | 26.97 | 29.83 | 53.32 | 48.46 |
| w.o. *Tucker Decomposition* | 72.02 | 26.73 | 29.65 | 51.41 | 48.22 |
| **GeneLLM-OLMoE** | **73.91** | **27.56** | **33.05** | **55.44** | **49.96** |

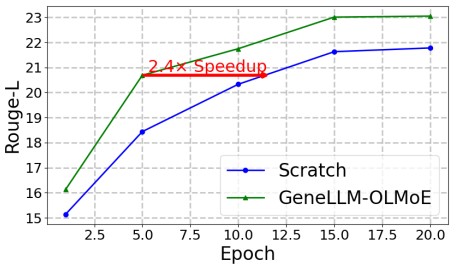 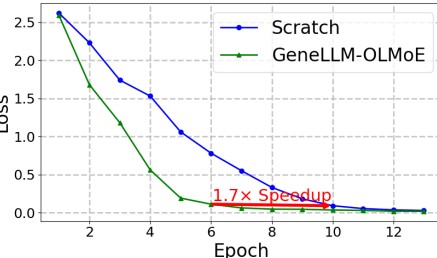

Figure 5: Rouge-L and loss curves on downstream dataset DollyEval.

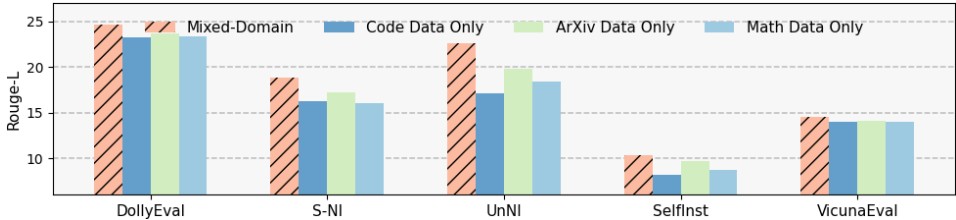

Figure 6: Impact of using different types of data for experts selection on SFT task performance of lightweight models. **Mixed-Domain** denotes the diverse corpus used by GeneLLM for experts selection, including (but not limited to) Wikipedia, GitHub, and arXiv.

pendix A.4). As shown in Table 4, two ablation variants are considered: w.o. *Experts Selection*, where experts are randomly chosen, and w.o. *Tucker Decomposition*, where expert parameters are simply averaged to initialize the lightweight model. In the full method (GeneLLM-OLMoE), 4 experts are selected via our strategy and compressed with Tucker decomposition. Across various language understanding benchmarks, GeneLLM-OLMoE consistently achieves superior performance compared to both ablation variants, highlighting the importance and effectiveness of expert selection and Tucker decomposition modules in GeneLLM.

To demonstrate that expert selection with diverse knowledge domains facilitates the extraction of more generalizable and transferable knowledge, we conduct experiments using different data types for experts selection (Figure 6). The results show that when experts are selected with mixed-domain data, the extracted learngene yields better initialization for lightweight models, enhancing performance and generalization on downstream SFT tasks. This is because experts consistently activated across heterogeneous data are more likely to encode domain-agnostic and common knowledge, broadly applicable across tasks. Hence, incorporating diverse knowledge domains in experts selection is crucial for improving the adaptability and generalization of lightweight models.

## 5 CONCLUSION

GeneLLM is proposed as a low-cost initialization method. By selectively inheriting cross-domain common knowledge from MoE-based LLMs, GeneLLM enables lightweight models to be quickly constructed and adapted for vertical-domain specialized sub-tasks. Extensive experiments show that GeneLLM allows small models to achieve over 80% of the source LLM's performance on certain tasks, while using only 1.25% of its parameters and requiring significantly less pre-training data. Compared with baselines that do not inherit the learngene, such as Scratch and Distillation, GeneLLM consistently delivers superior performance. The low cost, fast convergence, and strong adaptability of our initialized lightweight models across diverse sub-tasks highlight the practicality and effectiveness of GeneLLM in building lightweight domain-specialized models.

## ETHICS STATEMENT

This work does not involve human subjects, personal data, or sensitive information. The datasets we used are publicly available and widely adopted in prior research. We have carefully ensured that our methods and experiments comply with ethical standards and do not introduce additional risks related to privacy, fairness, or security. Our contributions focus on methodological advancements and do not include applications that might directly cause harm or raise safety concerns. We are committed to releasing our code and models in a transparent and reproducible manner, following best practices for research integrity. All authors have read and acknowledged the Code of Ethics during the submission process.

## REPRODUCIBILITY STATEMENT

We have taken several steps to ensure the reproducibility of our work. A detailed description of our proposed method, model architecture, and training procedures is provided in Section 3 and Section 4. All hyperparameters and implementation details are reported in Appendix A.2. The datasets used in our experiments are publicly available. To further facilitate reproducibility, we release an anonymous link to the source code and scripts for training and evaluation as supplementary material. Together, these resources enable researchers to fully reproduce our results.

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

# A APPENDIX

## A.1 EXPERTS FREQUENCY IN OLMOE-7B

In the Introduction 1, we present the activation frequency of experts in the FFN layer of Layer 15 in the OLMoE-7B model across different domains (Github, Tulu, and Arxiv). Here, we provide additional visualizations. Figure 7 shows the activation frequencies of experts in Layer 0 and Layer 7 of the OLMoE-7B model across various domains. It can be observed that different experts within the same layer exhibit similar activation trends across these domains. For example, Expert 0, 21, and 40 in Layer 0, as well as Expert 2, 4, 17, 35, and 58 in Layer 7, all maintain high activation across multiple domains. This phenomenon indicates that MoE architectures inherently suffer from load imbalance among experts. Experts that are consistently activated across domains likely encode knowledge shared by multiple domains. We consider these experts to contain common knowledge that can be applied to a variety of tasks.

## A.2 EXPERIMENTAL DETAILS

We set the embedding dimension of the lightweight models to 2048, with an intermediate size of 1024 in the FFN layers. To evaluate models of different scales, we experiment with 8, 12, and 16 layers lightweight models. Since we extract learngene from each block of the source LLM, the learngene is block-wise. Each model inherits learngene from the corresponding blocks of the source LLM, for example, the 8-layer lightweight model inherits the first 8 shallow blocks of learngene from the LLM for initialization. Then we pre-train all baseline models. For a fair comparison, all baselines undergo this pre-training with the same amount of tokens. After pre-training, the baselines are further fine-tuned on the aforementioned SFT datasets for evaluation.

For fine-tuning on the BoolQ, Hellaswag, MMLU, PIQA, and WinoGrande language understanding datasets, all baselines are trained for only 3 epochs. For evaluating generative capabilities of models, we follow the MiniLLM setting: models are first trained on the Dolly dataset for 100 epochs, and then evaluated on the DollyEval, S-NI, UnNI, SelfInst, and VicunaEval datasets.

**Datasets and code.** We use the RedPajama-V2 as our pre-training data, and the fine-tuning datasets for downstream tasks are presented in Section 4. As for the source code, we provide an anonymous link, check https://anonymous.4open.science/r/GeneLLM-main-0EB3/.

**Software Environment.** All experiments were conducted using PyTorch 2.1 with CUDA version 12.1 and 4×H100 GPUs.

**Model Configuration.** The LLMs are OLMoE-7B and DeepSeekMoE-16B, and the target lightweight models are Llama models (dense) with different size.

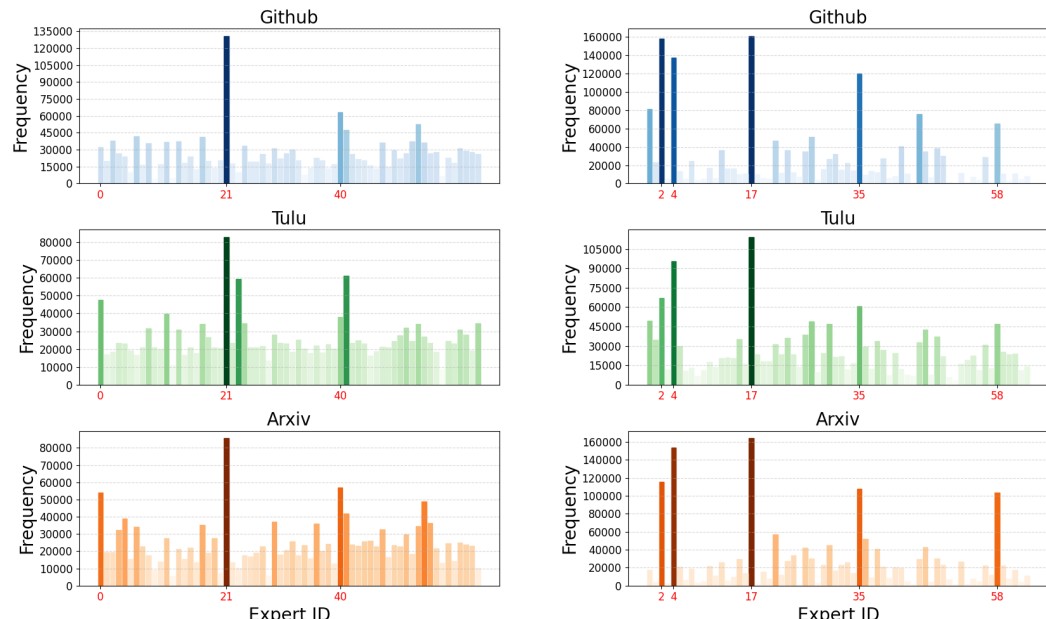

Figure 7: Expert activation frequencies of layer 0 (left) and layer 7 (right) in OLMoE-7B across different domains.

**Baseline Design Details.** In addition to the baselines based on GeneLLM (**GeneLLM-OLMoE** and **GeneLLM-DeepSeek**), we also include **Scratch** and **Distillation** as comparative baselines. Specifically, **Scratch** refers to a model that is randomly initialized, pre-trained, and then fine-tuned on downstream tasks. **Distillation** denotes a model that undergoes pre-training with distillation from the LLM, followed by fine-tuning on downstream tasks. To ensure the fairness of our comparisons, all baselines are configured with identical model parameter sizes and are pre-trained on datasets of the same scale.

**Training Hyperparameters.** The models are pre-trained using AdamW with an initial learning rate of 4e-4, batch size of 64, and a total of 20k-40k pre-training steps. During fine-tuning, we set the learning rate to 3e-4, and batch size to 8.

### A.3 MORE RESULTS WITH DIFFERENT AMOUNT OF PRE-TRAINING DATA

In this section, we present the fine-tuning results of various baselines on downstream datasets after pre-training with different amounts of data in Table 5 and Table 6. It can be observed that GeneLLM consistently outperforms other baselines across different tasks, demonstrating a significant advantage.

### A.4 ABLATION STUDY ON TUCKER RANK SETTINGS

In this section, we analyze how preserving different rank values along each dimension in the Tucker decomposition affects the experimental results. For the target lightweight model with a 2048-dimensional embedding and an intermediate size of 1024 in the FFN, we adopt two rank settings, (1024, 512) and (512, 256), corresponding to retaining $\frac{1}{2}$ and $\frac{1}{4}$ of the original parameter matrix size, respectively. As shown in Table 7, the three values in each rank setting correspond to $R_1$, $R_2$, and $R_3$ in Equation 5. From Table 7, we observe that, under the same number of selected experts, no particular rank setting shows a significant advantage over the other, suggesting that GeneLLM is robust to variations in rank settings within a reasonable range. Additionally, for the same rank setting, increasing the number of experts generally improves the performance of GeneLLM. However, this trend only holds within a certain range; if the number of Top-K selected experts becomes too large, the increased redundancy information in the experts can negatively impact the results.

Table 5: Results of models with different scales on model generation benchmarks (2B tokens pre-trained).

| Model | Baseline | DollyEval | S-NI | UnNI | SelfInst | VicunaEval |
|---|---|---|---|---|---|---|
| 8 Layers 391M | Scratch | 21.95 | 15.46 | 19.33 | 8.71 | 13.51 |
| | Distillation | 21.67 | 15.47 | 17.94 | 7.58 | 13.24 |
| | GeneLLM-OLMoE | 22.64 | **16.28** | 18.93 | **10.09** | 13.83 |
| | GeneLLM-DeepSeek | **22.66** | **16.28** | **20.17** | 9.35 | **14.60** |
| 12 Layers 480M | Scratch | 22.39 | 14.39 | 19.28 | 8.65 | 13.81 |
| | Distillation | 21.31 | 14.33 | **21.47** | 8.88 | 13.91 |
| | GeneLLM-OLMoE | **23.36** | **16.44** | 19.79 | **9.44** | 14.01 |
| | GeneLLM-DeepSeek | 22.76 | 14.32 | 19.06 | 9.38 | **14.16** |
| 16 Layers 570M | Scratch | 22.97 | 16.74 | 19.75 | 9.17 | 13.35 |
| | Distillation | 22.91 | 15.84 | 18.85 | 8.77 | 13.98 |
| | GeneLLM-OLMoE | **23.84** | **17.50** | **21.97** | 9.71 | **14.30** |
| | GeneLLM-DeepSeek | 23.31 | 16.71 | 20.89 | **10.26** | 13.24 |

Table 6: Results of models with different scales on model generation benchmarks (4B tokens pre-trained).

| Model | Baseline | DollyEval | S-NI | UnNI | SelfInst | VicunaEval |
|---|---|---|---|---|---|---|
| 8 Layers 391M | Scratch | 22.08 | 16.52 | 19.82 | 8.73 | 13.13 |
| | Distillation | 21.90 | 15.47 | 18.67 | 8.33 | 13.14 |
| | GeneLLM-OLMoE | 22.66 | **17.39** | **20.18** | **9.59** | **14.33** |
| | GeneLLM-DeepSeek | **22.70** | 17.04 | 19.74 | 8.29 | 13.77 |
| 12 Layers 480M | Scratch | 22.84 | 16.25 | 19.57 | 8.96 | 14.87 |
| | Distillation | 22.73 | 16.62 | 19.48 | 9.20 | 14.20 |
| | GeneLLM-OLMoE | 23.49 | **17.95** | **22.08** | 9.48 | **15.53** |
| | GeneLLM-DeepSeek | **23.76** | 17.40 | 20.24 | **10.43** | 14.79 |
| 16 Layers 570M | Scratch | 22.78 | 15.68 | 19.65 | 9.53 | 14.05 |
| | Distillation | 22.09 | 16.06 | 20.27 | 10.20 | 14.09 |
| | GeneLLM-OLMoE | 23.52 | **18.79** | **23.16** | 10.38 | **14.54** |
| | GeneLLM-DeepSeek | **23.90** | 17.30 | 21.93 | 11.05 | 14.19 |

## A.5 THE LEARNGENE CAN BE ADAPTED TO LIGHTWEIGHT MODELS OF ANY DIMENSION.

In Section 3.3, we introduce the case where the parameter matrices of the LLM and target small models have the same size. In this section, we describe how to adjust the size of the higher-order tensor $Z$ when the target lightweight model has an embedding dimension and intermediate size that do not match those of the LLM. This adjustment ensures that the learngene extracted and reconstructed from $Z$ can be properly adapted to any target size of the model.

To adapt the learngene to the model of any dimension, we first adjust the size of tensor $Z \in \mathbb{R}^{d_1 \times d_2 \times n}$ to $Z \in \mathbb{R}^{d_1' \times d_2' \times n}$, where $d_1'$ and $d_2'$ fit the dimensions of the parameter matrix in the target lightweight model. Here we give the process:

Table 7: Results of target lightweight models with different scales on language understanding benchmarks.

| Top-K Selected | #Rank Setting | BoolQ | Hellaswag | MMLU | PIQA | WinoGrande |
|---|---|---|---|---|---|---|
| 2 experts | (256, 512, 2) | 73.31 | 27.34 | 29.78 | 52.07 | 48.70 |
| | (512, 1024, 2) | 72.05 | 27.40 | 30.57 | 52.94 | 48.86 |
| 4 experts | (256, 512, 4) | 73.91 | 27.56 | 33.05 | 55.44 | 49.96 |
| | (512, 1024, 4) | 71.90 | 28.23 | 30.57 | 53.97 | 49.64 |
| 8 experts | (256, 512, 8) | 73.06 | 28.07 | 29.78 | 52.47 | 50.04 |
| | (512, 1024, 8) | 73.00 | 27.66 | 30.70 | 53.90 | 52.33 |

$$||Z_{i,:,:}||_F = (\sum_{j=1}^{d_2} \sum_{k=1}^{n} Z_{i,j,k}{}^2)^{\frac{1}{2}}, \quad \forall i \in \{1, \ldots, d_1\}, \tag{7}$$

$$S_r = arg \max_{|S|=d_1'} \sum_{i \in S} ||Z_{i,:,:}||_F, \tag{8}$$

$$||Z_{:,j,:}||_F = (\sum_{i \in S_r} \sum_{k=1}^{n} Z_{i,j,k}{}^2)^{\frac{1}{2}}, \quad \forall j \in \{1, \ldots, d_2\}, \tag{9}$$

$$S_c = arg \max_{|S|=d_2'} \sum_{j \in S} ||Z_{:,j,:}||_F, \tag{10}$$

$$\tilde{Z} = Z[S_r, \ S_c, \ :], \tag{11}$$

where $||Z_{i,:,:}||_F$ and $||Z_{:,j,:}||_F$ represent the norms of tensor $Z$ along its rows and columns, respectively. $S_r$ and $S_c$ denote the indices of the top $d_1'$ rows and top $d_2'$ columns with the highest norm values in the row and column directions. Finally, the resized tensor $\tilde{Z}$ is obtained by retaining only the rows in $S_r$ and columns in $S_c$, resulting in a tensor of dimensions $d_1' \times d_2' \times n$.

### A.6 ANALYSIS OF TUCKER DECOMPOSITION

In this section, we analyze the theoretical rationality behind using Tucker decomposition to extract shared knowledge among experts for refining the learngene. To identify shared knowledge across a group of expert parameter matrices, we first stack them into a third-order tensor and apply Tucker decomposition to extract compact representations along each dimension. Formally, given a set of matrices $\{X_i \in \mathbb{R}^{d_1 \times d_2}\}_{i=1}^{n}$, we construct a tensor $\mathcal{X} \in \mathbb{R}^{d_1 \times d_2 \times n}$ where $\mathcal{X}(:,:,i) = X_i$. This tensor captures both the internal structure of each matrix and the relationships across matrices.

We then perform Tucker decomposition:

$$\mathcal{X} \approx \mathcal{G} \ \times_1 \ U_1 \ \times_2 \ U_2 \ \times_3 \ \bar{U}_3, \tag{12}$$

where $\mathcal{G}$ is a compact core tensor and $U_1$, $U_2$, $U_3$ are orthogonal factor matrices capturing the most important components in each mode. By retaining only the top principal components (i.e., low-rank approximation), Tucker decomposition filters out high-frequency noise or task-specific variance while preserving the dominant, shared structures that are consistently present across the matrices.

Intuitively, if each matrix $X_i$ can be viewed as a combination of shared content $C$ and an instance-specific perturbation $\theta_i$, i.e., $X_i = C + \theta_i$, then stacking them as:

$$\mathcal{X} = \mathcal{C} + \Theta, \quad \mathcal{C}(:,:,i) = C, \ \Theta(:,:,i) = \theta_i. \tag{13}$$

Components with lower energy typically correspond to minor variations or idiosyncrasies specific to individual matrices, including random noise. By discarding these less significant components, the decomposition effectively denoises the representation and emphasizes structural coherence. If the matrices exhibit common patterns, such as similar structural characteristics or activation behaviors,

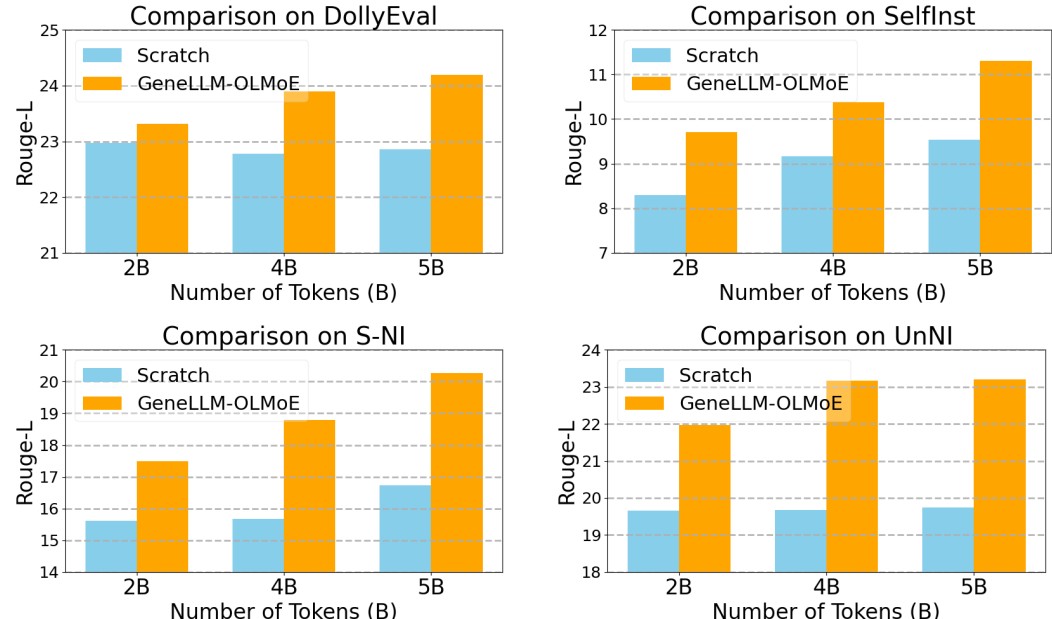

Figure 8: Comparison of fine-tuning performance between 16-layer GeneLLM-OLMoE and Scratch under varying pre-training token budgets

these shared structures tend to manifest as high-energy components in the tensor. Tucker decomposition captures these components through the leading principal directions, allowing us to reconstruct the tensor while retaining information that is commonly embedded across matrices.

Since the decomposition optimizes for the overall reconstruction error, the resulting low-rank representation preferentially captures the common structure shared across all expert matrices while suppressing task-specific noise or deviations. This enables us to distill domain-general knowledge encoded in the experts into a compact and reusable module, the learngene, which can be used to initialize target lightweight models with improved generalization and data efficiency.

## A.7 MORE RESULTS ON THE DATA EFFICIENCY OF GENELLM

We have previously introduced in Section 4.2 of the experiments a comparison between GeneLLM and Scratch under varying amounts of pre-training data, demonstrating that GeneLLM can achieve better performance than Scratch with significantly less data—saving even over 5× the pre-training cost. In this section, we present additional results to further highlight the data efficiency of GeneLLM. Specifically, we compare the downstream task performance of GeneLLM-OLMoE and Scratch after pre-training on 2B to 5B tokens across multiple datasets (DollyEval, SelfInst, S-NI and UnNI).

As shown in Figure 8, GeneLLM consistently outperforms Scratch across all tasks. Notably, Scratch pre-trained with 5B tokens still fails to surpass GeneLLM-OLMoE pre-trained with only 2B tokens. This further demonstrates the significant data efficiency of GeneLLM, highlighting its ability to greatly reduce the amount of required pre-training data and associated training cost.

## A.8 LIMITATIONS AND DISCUSSIONS

Although the Learngene framework offers a promising approach for reusing knowledge from large MoE-based models through efficient inheritance, there remain several aspects worth further discussion.

First, although GeneLLM avoids the computational overhead of distillation by relying solely on forward inference, it still assumes access to a large language model for the extraction phase. This could pose a challenge for extremely resource-constrained environments, though the cost is significantly lower than full-scale fine-tuning or distillation.

Second, while we demonstrate strong results across multiple tasks, GeneLLM currently adopts a uniform initialization strategy for all downstream tasks. Future work may explore task-adaptive initialization or hybrid approaches that combine learngene with lightweight adaptation techniques (e.g., LoRA or adapters) to further improve transferability.

Overall, we believe Learngene offers a compelling starting point for efficient model initialization, and we hope that it inspires future research on more flexible, modular, and reusable frameworks.

### A.9 BROADER IMPACTS

The development of more efficient and capable small language models has notable societal implications. By extracting knowledge from larger models and leveraging flexible initialization methods like Learngene, GeneLLM enables the creation of lightweight models that retain strong generalization and generation abilities with significantly reduced computational cost. This democratizes access to advanced AI systems, allowing individuals, research institutions, and enterprises with limited resources to benefit from high-quality language models.

Moreover, smaller and more efficient models contribute to the reduction of energy consumption and carbon footprint associated with large-scale AI deployment, aligning with global efforts towards sustainable AI practices.

Ultimately, we believe that advancing small model capabilities is a step towards more inclusive, efficient, and environmentally responsible AI technologies, provided that ethical guidelines and appropriate safeguards are continuously maintained.

### A.10 THE USE OF LLMS

In this paper, we only employ large language models (LLMs) for grammatical correction and language polishing of the manuscript. The use of LLMs does not involve the core ideas, methodological design, experimental implementation, or result analysis of the paper. We affirm that all methods and code, as well as the important equations and figures presented in the paper, are entirely written and produced by the authors. We declare that all the generated content of LLMs has been reviewed by the authors.

