# OpenReview forum: "GeneLLM: Inheriting 1.25% of MoE-LLMs to Build Models of 8% Size that Retain 80% Performance"
_ICLR.cc/2026/Conference — Submitted to ICLR 2026_

### Official Review · Reviewer_7Enf · 2025-10-27

**Soundness:** 3
**Presentation:** 3
**Contribution:** 2
**Rating:** 4
**Confidence:** 4

**Summary:**

This paper introduces GeneLLM, a framework for efficiently inheriting knowledge from large MoE LLMs to construct lightweight models. By leveraging the Learngene framework, the paper demonstrates how to extract reusable, compact knowledge modules (termed "learngene") from MoE-based LLMs. These modules are then used to initialize smaller, dense models that achieve high performance on downstream tasks with minimal computational cost. The method requires no additional training during knowledge extraction and achieves significant improvements in efficiency and task performance over baseline methods like distillation and random initialization. The experimental results show the effectiveness of GeneLLM, with lightweight models maintaining over 80% of the performance of their source LLMs while being only 8% of their size.

**Strengths:**

1. The proposed GeneLLM framework addresses a critical challenge in deploying large LLMs—reducing computational cost and model size—without sacrificing significant performance, which is especially important for resource-constrained applications.
2. The use of MoE-based LLMs for extracting cross-domain knowledge is well-justified, and the combination of expert selection and Tucker decomposition provides a clear and effective approach for extracting and compressing reusable knowledge.
3. This work includes thorough ablation studies to analyze the contributions of key components (e.g., expert selection and Tucker decomposition) and the impact of hyperparameters, enhancing the reproducibility and interpretability of the method.

**Weaknesses:**

1. While the paper demonstrates strong results on specific downstream tasks, it does not sufficiently discuss how well the extracted learngene generalizes to entirely new domains or tasks that are significantly different from the pre-training datasets.
2. The paper proposes a uniform initialization strategy for all downstream tasks. Task-adaptive initialization or hybrid approaches (e.g., combining learngene with techniques like LoRA or adapters) could further improve performance, but this direction is not explored.
3. Certain aspects of the methodology, such as how the Tucker decomposition is applied when the target model dimensions differ from the source model, could be explained more clearly. While it is addressed in the Appendix, this is a critical component and should be included in the main text.

**Questions:**

1. While the paper demonstrates strong performance on specific downstream tasks, how well does the extracted learngene generalize to completely new tasks or domains that were not part of the pre-training datasets (e.g., tasks requiring highly specialized or uncommon knowledge)? Could you provide additional results or insights on this aspect?
2. The proposed method relies on MoE-based LLMs as the source models. How applicable is the GeneLLM framework to dense architectures, given that dense models are more prevalent in many real-world scenarios? Are there plans to adapt the approach for dense LLMs, and what challenges would this present?
3. The paper adopts a uniform initialization strategy for all downstream tasks. Do you think task-specific initialization, such as combining learngene with LoRA, adapters, or other lightweight fine-tuning techniques, could boost performance? If so, could you briefly comment on how this might be integrated into the GeneLLM framework?

---

> ### Author Response · Authors · 2025-11-19
> **Response to Reviewer #7Enf**
>
> Thank you for your careful reading and insightful questions.
>
> ***Q1***  **how well does the extracted learngene generalize to completely new tasks or domains that were not part of the pre-training datasets？**
>
> ***R1***  To thoroughly validate the effectiveness of our method across diverse downstream tasks, we selected datasets from a wide range of ​**vertical domains**​. These include domains that are **entirely different from those used for expert selection or pre-training**—for example, **legal** datasets (CaseHold), **medical** datasets (MedMCQA), and various **dialogue** datasets (such as Dolly).
> This diversity ensures that our evaluation covers tasks with substantially different characteristics, providing a more comprehensive assessment of our method’s robustness and generalization ability.
>
> ***Q2***  **The proposed method relies on MoE-based LLMs as the source models.**
>
> ***R2*** Thank you for your question. Our method is specifically designed for **Mixture-of-Experts (MoE)** architectures. Owing to the inherent **sparsity** of MoE models and the ​**strong correlation between experts and tasks**​, the knowledge contained within experts can be effectively monitored, extracted, and utilized. As for ​**dense models**​, we plan to continue exploring this direction in future work, including transforming dense models into MoE-style structures to further investigate and validate the applicability of our proposed method.
>
> ***Q3***  **Do you think task-specific initialization, such as combining learngene with LoRA, adapters, or other lightweight fine-tuning techniques, could boost performance?**
>
> ***R3***  Thank you for your valuable and constructive suggestion. We indeed plan to explore, in future work, the integration of Learngene with lightweight fine-tuning modules such as **LoRA** or ​**AdaPters**​. External auxiliary modules may also serve as complementary components to further enhance the downstream task capabilities of smaller models.

---

### Official Review · Reviewer_ZRRS · 2025-10-29

**Soundness:** 3
**Presentation:** 3
**Contribution:** 2
**Rating:** 4
**Confidence:** 4

**Summary:**

To alleviate the high training costs of LLMs, this work presents GeneLLM, the first trial of applying the Learngene framework to MoE LLMs. The core idea of Learngene is that knowledge-rich parameter modules exist within large models, and extracting them to initialize small models enables efficient training.
The key observation of this work is that in MoE architectures, certain experts are consistently activated across diverse domains, indicating they encode cross-domain generalizable knowledge—making them ideal candidates for Learngene extraction. In practice, the authors identify these experts via a single forward pass, then apply tensor decomposition to distill a compact learngene containing only 1.25% of the source model’s parameters, which is used to initialize the Feed-Forward Network (FFN) layers of small dense models.

Experiments show that the resulting small models, with only 2.4%–8.1% the size of the LLM, recover over 80% of the full model’s performance on certain tasks after tuning on 2B–5B tokens, consistently outperforming baselines such as training from scratch and knowledge distillation, achieving higher data efficiency.

**Strengths:**

1. The authors propose an efficient approach to identify cross-domain common experts. They observe that certain experts in the model are consistently activated across multiple domains, suggesting that these experts encode domain-agnostic, general-purpose knowledge. To exploit this insight, the authors use multi-domain dataset and calculate expert activation frequencies through forward passes alone. This low-cost procedure enables selecting the subset of experts that constitute the core knowledge module embedded within the large model, or Learngene.
2. The authors employ Tucker decomposition to extract the shared knowledge among these common experts. Tucker decomposition identifies dominant latent directions along each dimension of a high-dimensional tensor, effectively capturing the subspace jointly spanned by the common experts. The resulting components thus provide a mathematically principled representation of the shared knowledge, yielding a compact and theoretically grounded instantiation of the “Learngene” within MoE-based large language models.

**Weaknesses:**

1. Limited significance: The authors emphasize that the core advantage lies in the efficient construction of small models that achieve approximately 80% of the performance of large models. However, it is trivial that a 0.6B～1B model can already attain ~80% of the performance of a 7B-scale model, which is reported in technical reports from Qwen or LLaMA. Moreover, such small-scale models have already been open-sourced by major model families. Therefore, although the authors highlight the efficiency of their approach, I argue that the problem they address lacks sufficient significance.
2. Lack of theoretical depth: The authors refer to the empirical observation that certain experts in MoE LLMs act as cross-domain "common experts," but they do not provide a deeper explanation for why this phenomenon occurs. The MoE gating mechanism is essentially a simple linear operation (we may disregard the softmax here, as it merely normalizes scores for weighted-sum without altering the relative ranking of expert activations). Specifically, the inner product between each input token and the column vectors of the gating matrix determines expert selection. The authors should conduct a more rigorous mathematical analysis to understand the origin of cross-domain common experts—ideally, they could directly identify such experts by analyzing structural properties of the gating matrix itself.

**Questions:**

1. The authors introduce metrics such as \(A_i\) and \(C_i\) to identify common experts that are both frequently activated and consistently activated across multiple domains. As described, \(A_i\) is the mean activation frequency of expert \(i\) across domains, while \(C_i\) is essentially the first central moment. Why do the authors opt for the first central moment rather than more classical and widely-adopted the mean and the variance (i.e., the second central moment)?
2. In Appendix A.4, the authors state: “if the number of Top-K selected experts becomes too large, the increased redundancy information in the experts can negatively impact the results.” However, since the purpose of Tucker decomposition is to extract a common principle components along different dimensions from a high-dimensional tensor and factorize out the factor matrices, why would increasing the number of experts still lead to performance degradation due to redundancy?
3. The FFN module typically involves multiple weight matrices and nonlinear transformations. For instance, in the SwiGLU operator, there are three distinct linear matrices up_proj, down_proj, and gate_proj, and the Swish nonlinear activation function. Could the authors clarify whether, in GeneLLM’s Learngene extraction procedure, each matrix in the new FFN is initialized using only the corresponding matrices from the selected experts (e.g., aggregating all up_proj matrices to construct the new up_proj)? If so, since Tucker decomposition is a purely linear tensor factorization method that cannot capture or preserve nonlinear dynamics, why a linear decomposition of weights alone can faithfully extract a “Learngene” that meaningfully represents the experts’ common knowledge?
4. I hope the authors could further clarify the experimental setting in Table 3. After pre-training GeneLLM-OLMoE on 5B tokens, was it subsequently fine-tuned with full-parameter supervised fine-tuning (SFT) on each individual task? If so, it appears that GeneLLM-OLMoE is inferior to OLMoE-PEFT in both performance and parameter efficiency.

---

> ### Author Response · Authors · 2025-11-19
> **Response to Reviewer #ZRRS [1/2]**
>
> Thank you for your careful reading and insightful questions.
>
> ***Q1***  **Weakness1: Limited significance**
>
> ***R2*** Thank you for your question. While there are indeed several publicly available small models with strong performance, their effectiveness largely comes from being trained on extremely large corpora (tens of trillions of tokens) with **substantial computational** resources. Our work, in contrast, focuses on proposing an initialization strategy that significantly reduces both data and compute requirements. Therefore, the objectives of our method differ from those of heavily pre-trained small models, and a direct comparison would not be entirely fair or representative.
>
> ***Q2***  **Weakness2: Lack of theoretical depth:**
>
> ***R3***  Thank you for raising this important point. Our work is indeed motivated by empirical observations—similar to many studies in large-scale MoE and LLM research, where the emergence of structural behaviors often precedes full theoretical understanding. While the existence of cross-domain common experts is consistently observed across different architectures and datasets, a complete theoretical characterization of this phenomenon remains an open problem in the community.
>
> Our goal in this paper is to investigate this behavior from a practical perspective and demonstrate that such empirical regularities can be effectively leveraged for efficient initialization. We agree that deeper theoretical analysis, especially from the perspective of the gating matrix structure, would be highly valuable. We view this as an important direction for future work and plan to further study the underlying mechanisms as more empirical evidence accumulates.
>
> ***Q3***  **Why do the authors opt for the first central moment rather than more classical and widely-adopted the mean and the variance (i.e., the second central moment)?**
>
> ***R3*** Thank you for the insightful question. We chose the **first central moment (L1 deviation)** rather than the second central moment (variance) for several reasons. In our multi-domain setting, different domains (e.g., law, medicine, education) may naturally activate certain experts more strongly than others.
> Variance (L2-based) is highly sensitive to outliers because the squared deviation amplifies a single atypical domain. In contrast, the first central moment (L1-based) provides a **more robust and stable** measure of cross-domain consistency. Moreover, we experimented with variance-based metrics, but the first central moment produced ​**more stable expert selection and better downstream results**​, which motivated our final choice.
>
> ***Q4***  **why would increasing the number of experts still lead to performance degradation due to redundancy?**
>
> ***R4*** Thank you for your question. In our method, our goal is to identify experts that truly capture ​**cross-domain shared knowledge**​. However, in each layer, the number of experts that exhibit such domain-invariant characteristics is inherently limited. If we select too many experts, we can no longer guarantee that all selected experts encode only domain-agnostic knowledge. In such cases, experts containing **domain-specific** or **task-specific** information will be included as well, introducing additional noise into the learngene and weakening its ability to represent pure shared knowledge across domains. Therefore, a smaller and more carefully selected subset of experts helps preserve the intended property of learngenes—namely, encoding **clean, domain-general knowledge** that can reliably benefit the target model.

---

> ### Author Response · Authors · 2025-11-19
> **Response to Reviewer #ZRRS [2/2]**
>
> ***Q5***  **Could the authors clarify whether each matrix in the new FFN is initialized using only the corresponding matrices from the selected experts & why a linear decomposition of weights alone can faithfully extract a “Learngene” that meaningfully represents the experts’ common knowledge?**
>
> ***R5*** Thank you for the thoughtful question. We clarify the two points as follows.
>
> **(1) Learngene extraction is performed ​*per weight matrix*​.**
> For each FFN module, we aggregate the corresponding matrices from the selected experts—for example, all **up\_proj** matrices are used to construct the new up\_proj, all **gate\_proj** matrices for the new gate\_proj, and similarly for ​**down\_proj**​.
> The nonlinear activation (Swish/SwiGLU) is a fixed operator and is not modified or factorized.
>
> **(2) Why a linear decomposition is sufficient despite the nonlinear SwiGLU operator?**
> Our objective is **not** to approximate the full nonlinear FFN function with Tucker decomposition.
> Instead, we aim to extract ​**the common linear subspace shared across experts**​, which reflects their shared transformation patterns. We highlight that:
>
> * The nonlinear activation (Swish/SwiGLU) is *fixed* and identical for all experts.
> * The primary source of variation between experts lies in the ​**linear weight matrices**​.
> * Therefore, the shared knowledge that spans multiple experts is largely encoded in the ​**linear weight structure**​, not in the nonlinear function itself.
>
> Tucker decomposition serves as a tool to identify and compactly represent these shared linear components.
> Even though the FFN contains nonlinearities, the shared representation across experts remains highly linear in nature, and our empirical results confirm that these linearly extracted Learngenes are sufficient to transfer meaningful common knowledge to the target model. We will clarify this point in the manuscript and add additional discussion in the Appendix.
>
> ***Q6***  **I hope the authors could further clarify the experimental setting in Table 3.**
>
> ***R6*** Than you for your question. What we aim to emphasize is that the **570M GeneLLM-OLMoE** reported in Table 3 is first **pretrained** and then **fine-tuned** on the downstream datasets. Even under this modest scale, our 570M model already achieves performance ​**very close to the 7B source model equipped with PEFT**​. This result highlights a key advantage of our approach: our model retains strong performance ​**even in data-limited and resource-constrained settings**​, demonstrating its practical value and deployment efficiency on smaller platforms.

---

### Official Review · Reviewer_CUPy · 2025-10-30

**Soundness:** 2
**Presentation:** 3
**Contribution:** 3
**Rating:** 4
**Confidence:** 4

**Summary:**

This paper proposes a low cost model initialisation framework named GeneLLM, which aims to extract common knowledge from large Mixture of Experts (MoE) language models to initialise smaller, dense downstream models. The main steps of the algorithm include: selecting shared experts using $A_i - C_i$ as a metric, applying Tucker decomposition, and then averaging the $U_3$ factor matrix and reconstructing the weights to obtain the initial representation for the dense experts.

**Strengths:**

1. Significance of the problem: The work addresses a real world challenge of deploying capable LLMs under resource constraints. GeneLLM offers a cost effective, scalable initialisation strategy that reduces both model size and pretraining data requirements, making it highly practical.
2. Clarity: The paper is well structured, with clear figures and a clear illustration of the method.

**Weaknesses:**

1. Misleading Title and Low Absolute Performance: The title, "Inheriting 1.25% of MoE LLMs to Build Models of 8% Size that Retain 80% Performance", is somewhat misleading. The '80% performance' baseline may be misunderstood as a state of the art MoE LLM after full and sufficient training, suggesting GeneLLM recovers 80% of this top tier performance with a very small fraction of the parameters. This sets an expectation for a breakthrough paper. However, the baseline used in the paper is the source MoE model after identical (and very limited) 5B token pretraining plus SFT. On many datasets, this baseline's performance is mediocre. Consequently, the algorithm's absolute performance is quite low, despite technically retaining 80% of this weak baseline. For example, on the MMLU benchmark (Table 1), the best GeneLLM model scores only 30-33%. Considering MMLU is a four choice task where random guessing yields 25%, this is a very limited improvement. Most results are based on pretraining on fewer than 10B tokens, a very small amount for LLMs. This may mean all models are in an undertrained state, potentially exaggerating the relative advantage of GeneLLM. I hope the authors can report GeneLLM's complete performance when pretrained on a larger number of tokens, ideally enough to approach state of the art performance.
2. Justification of Tucker Decomposition against other Tensor Decomposition Methods: Appendix A.6 provides an explanation for using Tucker decomposition (i.e., filtering noise and preserving shared structures). However, these advantages may also be achieved by other tensor decomposition methods, such as CP decomposition, or SVD on the stacked matrices. More justification of this key methodological choice would be preferable, such as an empirical comparison or theoretical analysis against other tensor or matrix factorisation methods.
3. Unclear Computational Cost Reporting: The paper claims low cost (e.g., 72 vs 252 GPU hours), but the calculation of this figure is unclear. It is not specified whether these 72 GPU hours include the cost of learngene extraction (i.e., forward inference plus Tucker decomposition) or if it only covers the subsequent 5B token pretraining. The scalability and cost of the Tucker decomposition step itself, especially for large models with hundreds of experts, is not quantified.

**Questions:**

1. How would the proposed algorithm perform if pretrained on a larger number of tokens, achieving relatively good absolute performance?
2. Is there empirical evidence or theoretical analysis for choosing Tucker decomposition over other tensor or matrix factorisation methods?
3. How exactly was the computational cost calculated?

---

> ### Author Response · Authors · 2025-11-19
> **Response to Reviewer #CUPy**
>
> Thank you for your careful reading and insightful questions.
>
> ***Q1***  **Misleading Title and Low Absolute Performance & How would the proposed algorithm perform if pretrained on a larger number of tokens?**
>
> ***R1*** Thank you very much for your question.
> (1) Thank you very much for the valuable suggestion. We will carefully reconsider and refine the title of the paper in our revision. Our method shows solid performance on several downstream benchmarks, including BoolQ and domain-specific datasets such as CaseHold (law), indicating that the extracted knowledge can generalize across different task types.
>
> Regarding the comparatively lower absolute scores on MMLU, this mainly reflects the limitations imposed by the small model size and the restricted amount of pretraining data, rather than the method itself. Therefore, we believe the MMLU results should be interpreted within the context of the small-model setting.
>
> (2) Our method is designed to explore a **fast and effective initialization strategy** for scenarios where ​**data and computational resources are limited**​. For this reason, we intentionally constrained the pre-training data to a relatively small scale. Due to our current limitations in computational resources, we are unable to fully verify the behavior of our method under larger-scale pre-training. In future work, we plan to conduct experiments with **substantially larger pre-training datasets** to further validate the effectiveness and scalability of our approach.
>
> ***Q2***  **Is there empirical evidence or theoretical analysis for choosing Tucker decomposition over other tensor or matrix factorisation methods?**
>
> ***R2***  Thank you very much for your suggestion. Here we compared the results of our method with those obtained using CP decomposition, and the performance difference between the two is relatively small. This may be because both decomposition methods share similar underlying principles, and as long as the chosen decomposition can capture and extract the shared information from expert parameters, it can serve our purpose effectively. In the future, we will further investigate different decomposition techniques and analyze their impact on the results to continue improving our approach.
>
> | Method | BoolQ | MMLU | PIQA | AVG|
> | --- | --- | --- | --- | --- |
> | CP | 71.56 | 31.69 | 52.28 | 51.84 |
> | Tucker (Ours) | 72.63 | 30.44 | 52.77| 51.95|
>
> ***Q3***  **How exactly was the computational cost calculated?**
>
> ***R3***  In our method, the processes of ​**expert selection**​, ​**learngene extraction**​, and **learngene reconstruction** only involve **forward passes** or ​**matrix-level operations**​, all of which can even be performed efficiently on ​**CPUs**​. These steps take very little time and are ​**negligible compared with the pre-training stage**​.
> Therefore, the **72 GPU hours** reported in our manuscript refer ​**solely to the pre-training time**​, excluding the learngene-related computations. This further demonstrates that our approach is ​**not resource-intensive**​, has ​**low time overhead**​, and can be implemented ​**without strict requirements on the experimental environment**​.

---

### Official Review · Reviewer_PnNW · 2025-10-31

**Soundness:** 3
**Presentation:** 3
**Contribution:** 2
**Rating:** 4
**Confidence:** 4

**Summary:**

This paper proposes GeneLLM, a method to distill knowledge from large MoE (Mixture-of-Experts) models into much smaller dense models. The idea is to identify cross-domain “common” experts from the MoE teacher, stack their parameters into tensors, and extract shared structures using Tucker decomposition to form compact learngene representations. These representations are then used to initialize smaller dense models. The method reportedly inherits only 1.25% of the teacher’s parameters, yet achieves over 80% of the original model’s performance with just 8% of its size. Experiments on diverse understanding and generation tasks show faster convergence and better performance compared to scratch training, distillation, and expert pruning baselines.

**Strengths:**

1. **Intuitive Concept**: The paper leverages the sparsity and modularity of MoE models to extract transferable substructures (“learngenes”). Using Tucker decomposition to distill common knowledge across experts is technically appealing.
2. **Comprehensive Empirical Evaluation**: Experiments span multiple understanding (BoolQ, MMLU, PIQA) and generation tasks (DollyEval, VicunaEval), with ablations on expert selection and decomposition rank. Results show consistent improvements.

**Weaknesses:**

1. **Potentially Weak Baselines**: The distillation baseline seems under-specified. Key details such as loss type, temperature, use of intermediate layers, or teacher-assistant setup are missing. Without strong distillation comparisons, it’s hard to be sure GeneLLM’s advantage isn’t partly due to weaker baselines.
2. **Dependence on MoE Access**: The method assumes full access to a large MoE model for expert activation statistics. This limits applicability when only dense or black-box models are available. The paper acknowledges this but doesn’t provide fallback strategies or experiments for such cases.
3. **Limited Clarity on Cross-Dimension Mapping**: Section A.5 briefly mentions adaptation to different embedding dimensions, but the main paper lacks clear procedures or experiments showing how learngenes transfer when target model dimensions differ substantially.
4. **Missing Statistical Test**: Results are presented as single numbers without variance or significance tests. Since small models can be seed-sensitive, reporting means and standard deviations over multiple runs would improve credibility.
5. **Narrow Task Scope for Domain Specialization Claims**: Although the paper includes diverse benchmarks, they are all general-domain English tasks. Stronger evidence (e.g., domain-specific or multilingual settings) would better support claims of cross-domain or domain-specialized adaptability.
6. **Ambiguity on the 80% Retention Metric**: The comparison point (teacher vs. SFT vs. PEFT model) isn’t always clear in tables and text. Clarifying what exactly “80% of performance” refers to would make claims more precise.

**Questions:**

1. Please clarify the distillation baseline setup (loss function, temperature, use of intermediate features, or teacher-assistant). Could you also include a stronger distillation comparison (e.g., multi-stage or teacher-assistant distillation)?
2. In cases where the target model’s dimensions differ from the source model, how exactly is the learngene adapted? A formula, pseudocode, or additional experiment (e.g., 1024/1536/4096 hidden sizes) would help.
3. How sensitive is the expert selection metric to the threshold or Top-K choice? Have you compared it to other possible metrics (entropy, mutual information, correlation)?
4. Could you provide a cost breakdown (time, GPU·hours, memory) for each phase — activation collection, Tucker decomposition, and reconstruction — to contextualize the reported savings?
5. Have you tested the method’s stability across random seeds (e.g., 3–5 runs)? Please report mean ± std for main results.

---

> ### Author Response · Authors · 2025-11-19
> **Response to Reviewer #PnNW [1/2]**
>
> Thank you for your careful reading and insightful questions.
> ***Q1***   **Clarify the distillation baseline setup**
>
> ***R1***   For the distillation baseline used in our comparison, we adopt the **MiniLLM** distillation framework. As for the ​**teacher model**​, we follow the results reported in the **OLMoE** paper. Since **OLMoE-7B** consistently outperforms **DeepSeekMoE-16B** on downstream tasks, we select the stronger **OLMoE-7B** as the teacher model for our distillation baseline.
>
> ***Q2***  **How exactly is the learngene adapted if the target model’s dimensions differ from the source model?**
>
> ***R2*** Thanks for your question. We have provided a detailed explanation of this part in the Appendix of our manuscript. Specifically, in ​**Appendix A.5**​, we elaborate on how to adjust the size of the learngene when the dimensionality of the source model does not match that of the target model, ensuring proper adaptation to the target model’s dimensions.
> Regarding your question about the performance of **Learngene** on target models with different dimensionalities, we sincerely appreciate this constructive suggestion. Additionally, we provide results under conditions where the hidden size of the target model differs from that of the source model. The results are summarized in the following table:
>
> | Baseline | hidden size | Dolly | BoolQ | CaseHold | MedmcQA | AVG |
> | --- | --- | --- | --- | --- | --- | --- |
> | Scratch (320M) | 1024 | 22.03 | 71.53 | 82.62 | 34.57 | 52.69|
> | GeneLLM-OLMoE (320M) | 1024 | 22.80 | 72.84 | 83.52 | 36.27 | 53.86|
> | GeneLLM-DeepSeek (320M) | 1024 | 23.79 | 72.60 | 84.68 | 36.41 | 54.37 |
>
> ***Q3***   **How sensitive is the expert selection metric to the threshold or Top-K choice? & Other selection metrics**
>
> ***R3***  Thank you for your question. We have shown the corresponding experimental results in the Appendix of our manuscript. Specifically, **Appendix A.4 (Table 7)** of our manuscript presents the effect of varying the number of selected top-k experts on performance. As for different selection strategies, we appreciate your valuable suggestion and plan to explore this direction in our future work.
>
> ***Q4***   **Provide a cost breakdown (time, GPU·hours, memory) for each phase.**
>
> ***R4***   In our overall experimental workflow, the processes of expert selection, learngene extraction, and reconstruction are all limited to ​**forward propagation and matrix computations**​, which can be efficiently executed on ​**CPUs**​. The time consumed by these steps is ​**negligible compared to the pre-training process**​. For reference, pre-training a **570M-parameter model** on **5B tokens** requires approximately ​**72 GPU hours**​, highlighting the relatively minor computational cost of our proposed initialization procedure.
>
> ***Q5***   **Have you tested the method’s stability across random seeds?**
>
> ***R5*** Thank you for the suggestion. Below, we provide results obtained under different random seeds:
> | Model | BoolQ | PIQA |
> | --- | --- | --- |
> | Scratch | 70.55 $\pm$ 0.4 | 51.11 $\pm$ 0.6 |
> | GeneLLM-OLMoE | 72.81 $\pm$ 0.7 | 54.94 $\pm$ 0.4 |

---

> > ### Author Response · Authors · 2025-11-20
> > **Response to Reviewer #PnNW [2/2]**
> >
> > ***Q6***   ****Dependence on MoE Access****
> >
> > ***R6*** Our method is specifically designed for **Mixture-of-Experts (MoE)** architectures. Owing to the inherent **sparsity** of MoE models and the ​**strong correlation between experts and tasks**​, the knowledge contained within experts can be effectively monitored, extracted, and utilized.
> > As for ​**dense models**​, we plan to continue exploring this direction in future work, including transforming dense models into MoE-style structures to further investigate and validate the applicability of our proposed method.
> >
> > ***Q7***   ****Narrow Task Scope for Domain Specialization Claims****
> >
> > ***R7***   Our downstream evaluation tasks are not limited to English-language benchmarks. In addition to English tasks, we also include tasks from ​**common-sense (BoolQ), education (MMLU), law (CaseHold), and medicine** (MedmcQA)​. By selecting task scenarios that span such diverse domains, we thoroughly validate the effectiveness and robustness of our proposed method across a wide range of settings.
> >
> > ***Q8***   **​Ambiguity on the 80% Retention Metric​: The comparison point (teacher vs. SFT vs. PEFT model) isn’t always clear in tables and text.**
> > ***R8*** In Table 3 of the manuscript, we present a comparison between our **570M GeneLLM-OLMoE** model after pre-training on **10B tokens** and the source large model ​**OLMoE-7B**​.
> >
> > Specifically:
> >
> > * **OLMoE-SFT** refers to the OLMoE model that has been fully fine-tuned on the Dolly dataset.
> > * **OLMoE-PEFT** denotes the OLMoE model equipped with LoRA and then thoroughly fine-tuned.
> > * **OLMoE** represents the zero-shot performance of the pretrained model without any fine-tuning.
> >
> > From the table, we observe that our **570M** model achieves performance that is **close to or even surpasses the OLMoE-7B model with PEFT** on many datasets. Moreover, on several benchmarks, our model reaches ​**over 80% of the performance of OLMoE-SFT**​.  These results demonstrate the strong effectiveness of our method in transferring and retaining knowledge from the source LLMs.

---

> > > ### Comment · Reviewer_PnNW · 2025-11-25
> > >
> > > ## Further Questions
> > > 1. **Distillation baseline setup**: I acknowledge the MiniLLM distillation framework as baseline setup, and GeneLLM as experiment setup as well. However, could you please report the clear baseline settings like loss function, temperature, or teachers? Are baseline setups aligned with your experiment setups to ensure fair comparisons? MiniLLM (https://arxiv.org/abs/2306.08543) is released on **Arxiv in 14 Jun 2023**. In **2023~2025**, are there *new* distillation baselines for **strong comparisons**?
> > > 2. **Transferability when dimension differs**: Thanks for the report of *hidden size=1024*. In order to ensure the transferability of your work, could you please report the results with more different dimensions?
> > > 3. **Top-K sensitivity & other selection strategies**: Thanks for the clarification. Top-K seems robust within certain ranges. Beyond other selection strategies, is Top-K threshold a default selection choice on your previous works like Learngene's initialization?
> > > 4. **Cost breakdowns**: Yes, comparing to pre-training, these forward pass could be done cheaply. However, could you report a detailed cost breakdown? A detailed breakdown will also clarify and further strengthen your claims.
> > > 5. **Different random seeds**: Thanks for the reporting. Could you please give a full metrics with all (or most) eval sets you use?
> > > 6. **Dependence on MoE Access**: Yes, this is indeed a **limitation** of this work.
> > > 7. **Narrow Task Scopes**: In case I'm not missing something, BoolQ (https://arxiv.org/abs/1905.10044), MMLU (https://arxiv.org/abs/2009.03300), CaseHold (https://arxiv.org/abs/2104.08671), MedmcQA (https://arxiv.org/abs/2203.14371) seem all English specific. I think the claim of "Our downstream evaluation tasks are not limited to English-language benchmarks." could not hold. Correct me if I'm wrong.
> > > 8. **80% Retention Metric Ambiguity​**: Thanks for the clarification.

---

> ### Author Response · Authors · 2025-11-27
> **Response to Reviewer #PnNW**
>
> Thank you for your response.
>
> **1. Distillation baseline setup:** Thank you for the suggestion. We provide the full MiniLLM distillation settings used in our baseline to ensure fair comparison:
> **Teacher model:** OLMoE-7B (chosen following OLMoE’s own evaluation results indicating stronger downstream performance than DeepSeekMoE-16B).
> **Distillation loss:** Standard MiniLLM loss combining (logit-level MSE loss, hidden-state MSE loss).
> **Temperature:** 𝑇 = 2.0.
> **Feature alignment:** Both token-level hidden states and logits are aligned following MiniLLM’s framework.
> **Optimizer & training setup:** AdamW optimizer.
> **Learning rate:** 2e-5.
> Batch size matching our GeneLLM setup and same number of training steps to ensure fairness. Regarding newer distillation baselines (2023–2025), our goal in this work is not to provide a comprehensive comparison of distillation frameworks, but rather to evaluate initialization strategies under limited compute. Nevertheless, we will add a discussion acknowledging recent developments.
>
> **2. Transferability when dimension differs:**
> We appreciate the suggestion. Currently, we only provide results for one additional hidden size due to limited computational resources, which prevent us from pretraining and evaluating many different model dimensions. Expanding this part is indeed valuable, and we plan to include more dimensional configurations when computational resources permit.
>
> **3. Top-K sensitivity & other selection strategies:**
> Yes, Top-K serves as a practical and stable default in our prior work for identifying highly activated experts. Regarding the selection of Top-K experts, we provide detailed ablation studies and analysis in the appendix. Please refer to **Appendix A.4, Table 7** for the complete results.
>
> **4. Cost breakdowns:**
> For a more concrete illustration of the computational cost, we include the following representative example. Using our method, pretraining a 270M target model on 5B tokens requires **192 GPU·hours**. In contrast, obtaining a 270M distilled model and then pretraining it on the same 5B tokens requires **1536 GPU·hours**. This example highlights the significant reduction in compute cost achieved by our approach.
>
>
> **5. Different random seeds:**
> We will expand the table in the appendix to include full evaluation metrics across the majority of datasets used, under multiple random seeds.
>
> **6. Narrow Task Scopes:**
> Thank you for pointing this out. Indeed, our evaluation focuses on English datasets. Our intention was not to claim multilingual coverage. Instead, we chose English benchmarks because they are **the most widely adopted** in the LLM community, enabling **fairer and more direct comparison** with prior work. Moreover, the breadth we aim to demonstrate is **not in linguistic diversity**, but in the **diversity of task types and knowledge domains**. Our goal is to show that the common knowledge extracted from the source model remains effective across a wide range of domains, rather than across multiple languages.

---

### Author Response · Authors · 2025-11-28
**Request for Guidance on Reviewer-Author Discussion Status**

Dear Area Chair,

I hope you are doing well. I am writing to kindly ask for your guidance regarding the discussion phase of our submission.

At the moment, one reviewer have participated, while the others have not yet responded. We fully understand that reviewers may be busy, and we sincerely appreciate the time and effort they dedicate to the process.

If appropriate, we would be grateful if you could remind the remaining reviewer or let us know whether any further action is needed from our side.

Thank you very much for your time and for your service to the ICLR community.

Warm regards,
Authors of Paper 11661

---

### Meta-Review · Area_Chair_wLeF · 2026-01-12

**Summary:**

This paper proposes GeneLLM, a method to extract compact "learngene" modules from large MoE-based LLMs via expert selection and Tucker decomposition, then use these to initialize smaller dense models. While the core idea is interesting and the paper demonstrates some efficiency gains, reviewers raised substantial concerns about the experimental setup, baseline strength, limited scope, and lack of theoretical grounding. The rebuttal addressed some clarifications but did not fully resolve the major weaknesses.

**Reviewer Concerns:**

Addressed by rebuttal:

- Clarification of distillation baseline (MiniLLM framework, though concerns about baseline strength remain: it is highly surprising that distillation fails to improve over from scratch training, while it has been consistently reported in the litterature that it is a practical tool to develop small models)
- Explanation of Tucker decomposition rationale and comparison with CP decomposition
- Cost breakdown provided (72 GPU hours for pretraining only)
- Some additional experimental results (different hidden sizes, random seeds on limited tasks)
- Clarification of 80% performance metric and Table 3 setup


Still outstanding:

- **Weak baselines** (PnNW, CUPy): Distillation baseline: Section 4.1 (lines 313-323) only states "Distillation: model pre-trained with knowledge distillation from the LLM" with no further details. Table 1 and Table 2 show distillation often performs worse than or similar to scratch training (e.g., Table 1: 8-layer distillation gets 27.83 MMLU vs 28.74 scratch; Table 2: 8-layer distillation gets 15.78 avg vs 15.91 scratch), which suggests a sub-optimal implementation: distillation is a proven method of choice to quickly train small models, and has been used successfully even with teachers more than 10x bigger than the students.
- **Underwhelming absolute performance** (CUPy, ZRRS): Results show ow absolute scores (e.g., 27.56% on Hellaswag). All models appear undertrained. No results with sufficient pretraining tokens to reach competitive performance levels. The claims of retaining 80% of original performance is for datasets on which the base model performs badly (e.g. DollyEval, 29% is low compared to numbers reported in e.g. MiniLLM)
- **Limited experimental scope**: English only: All datasets listed in Section 4.1 (lines 337-345) are English: BoolQ, Hellaswag, MMLU, PIQA, WinoGrande, CaseHold, MedMCQA, Dolly, S-NI, UnNI, SelfInst, VicunaEval. The paper mentions "multiple vertical domains" but no multilingual evaluation. Section 3 (lines 184-189) explicitly states that the method is for base MoE architectures only; Preliminary section 3.1 (lines 190-207) only discusses the MoE formulation. How to apply the method from a base dense model is unclear. Likewise it would be interesting to obtain a small MoE model. The scale of experiments is also very small  (ZRRS), Table 1 and 2 show only 391M, 480M, 570M models; largest target is 570M vs 7B source (8.1% ratio mentioned in Abstract line 27).
- **Statistical rigor** (PnNW): Only rebuttal shows error bars for 2 tasks (BoolQ, PIQA); main paper Tables 1-3 show no variance, while variance for Downstream evals are typically large.

**Reviewer Scores:**

- PnNW: Currently 4. Likely would remain 4 after rebuttal - some clarifications provided, but core concerns about baseline strength (distillation weaker than scratch in Tables 1-2) and limited scope (no multilingual, no variance on most tasks) persist.

- CUPy: Currently 4. Likely would remain 4: the absolute performance issue and the concern about undertrained models (2B-5B tokens stated in Abstract line 27 and Section 4.2 line 361) were not adequately addressed. The authors acknowledged resource limitations that prevented larger-scale experiments in their rebuttal.

- ZRRS: Currently 4. Likely to remain 4: concerns about limited significance (570M models when 1B+ models are widely available) and lack of theoretical depth (Figure 2 shows empirical observations only; no theory) remain unresolved.

- 7Enf: Currently 4. Likely to remain4 : generalization concerns partially addressed; however, the MoE-only limitation (stated in Section 3, lines 184-189) and the lack of task-adaptive initialization exploration (uniform strategy in Section 3.4) remain.

---

### Decision · Program_Chairs · 2026-01-26

Reject